**Peer**J

# The impact of maths support tutorials on mathematics confidence and academic performance in a cohort of HE Animal Science students

Nieky van Veggel and Jonathan Amory

School of Sport, Equine and Animal Science, Writtle College, Chelmsford, Essex, UK

## ABSTRACT

Students embarking on a bioscience degree course, such as Animal Science, often do not have sufficient experience in mathematics. However, mathematics forms an essential and integral part of any bioscience degree and is essential to enhance employability. This paper presents the findings of a project looking at the effect of mathematics tutorials on a cohort of first year animal science and management students. The results of a questionnaire, focus group discussions and academic performance analysis indicate that small group tutorials enhance students' confidence in maths and improve students' academic performance. Furthermore, student feedback on the tutorial programme provides a deeper insight into student experiences and the value students assign to the tutorials.

## INTRODUCTION

According to the 2010–2015 Strategic Plan of the Biotechnology and Biological Sciences Research Council (BBSRC), which funds bioscience research in the UK, there is an urgent need to raise the mathematical and computational skills of biologists at all levels due to the increasingly quantitative nature of the bioscience disciplines (*BBSRC, 2012*) and the trend in the workforce towards positions requiring higher levels of management expertise and problem-solving skills, many of which are mathematical in nature (*ACME, 2011*). In contrast to these developments, *Hodgen et al. (2010)* reported that the UK has the lowest participation of students in post-16 maths out of 24 OECD countries, the Royal Society reports that only 40% of students taking A level Biology also take A level Mathematics and reports published by the *Engineering Council (2000)* and by *Ramjan (2011)* confirm that this trend is not limited to the biosciences. A-levels (formally known as General Certificate of Education, Advanced level, contains no compulsory maths component) are subjects taught to 16–18 year old learners who have completed their General Certificate of Secondary Education (GCSE, ages 14–16 years old, contains a compulsory maths component). This leaves a gap between the knowledge and skills that are required for undergraduate bioscience degrees and the knowledge and skills with which new entrants to these degrees present. For example, *Tariq (2002)* reports that many entrants on a bioscience

Corresponding author
Nieky van Veggel,
Nieky.vanveggel@writtle.ac.uk

degree lack the skills that define a "numerate individual", even though most of them have at least a grade C (on a scale of A* to E, A* being highest) in GCSE maths, and *Tariq, Stevenson & Roper (2005)* describe that deficiencies in mathematics skills exist. *Tariq & Durrani (2009)* report that employers continue to voice concerns about the numeracy skills of their recruits and more recently *Koenig (2011)* reported that a general agreement exists amongst academic staff that a lack of mathematics knowledge, skill or confidence is preventing postgraduate bioscientists from becoming involved in interdisciplinary research.

One can wonder where this problem should be solved: at secondary level or at tertiary level? UK higher education teaching staff agree the GCSE and A-level curricula are no longer preparing students for a university education (*Browning & Sheffield, 2008*), with teachers no longer teaching skills, but teaching to syllabuses instead (*Julien & Barber, 2009*).

In order to address these issues numerous strategies to improve numeracy have been implemented by UK HE institutions. *Tariq (2002)* for example describes summer courses, diagnostic tests, "drop in surgeries" and encouraging the application of mental maths in order to improve numeracy, whereas *Miller & Walston (2010)* mentions the use of inter-disciplinary teams for teaching biosciences, *Tariq, Stevenson & Roper (2005)* adopt a case-study approach and *Ramjan (2011)* describes the use of contextualised diagnostic papers, all of which aim to place maths in a context that might provide more insight to the student.

The aim of this study was to investigate the effect of small group maths tutorials on the maths confidence and academic maths performance of first year undergraduate students enrolled on an animal science or animal management degree course. This paper describes the efficacy of use of such small-group mathematics tutorials and it investigates the possibilities of this type of mathematics support and the effect it has on the numeracy of a specific cohort of students.

## METHODS

This project was undertaken at Writtle College, a specialist land-based Higher Education institution in the Essex region. It consisted of three parts: a survey questioning students about their mathematics confidence, a set of focus group discussions to provide in-depth information on student motivation and an analysis of academic performance in modules with mathematical content.

The study population consisted of the 2011–2012 cohort of students ($N = 101$) enrolled on the first year of an Animal Management or Animal Science programme on either FdSc or BSc (Hons) level. Students were alerted to the tutorials by academic staff and encouraged to attend if their score on a pre-tutorial diagnostic test was below 4/10. However, student participation in the tutorials was entirely voluntary and the tutorials were optional. The pre-entry qualifications of these students (Table 1) were mainly on the Framework of Higher Education Qualifications (FHEQ) level 3, but varied in type of qualification (more vocational or more academic). In this framework, level 1 is the entry level which equates to completion of GCSE level studies and level 8 is the highest level

**Table 1 Analyses of the previous mathematics experience of students.** Analyses of the previous mathematics experience of students (compulsory only or post-compulsory) in relation to their tutorial attendance (attended or not attended), course subject (animal management or animal science) and course level (FdSc or BSc (Hons)).

| | Mathematics experience | | $\chi^2$ | $P$ |
|---|---|---|---|---|
| | Compulsory N (%) | Post-compulsory N (%) | | |
| *Tutorial attendance* | | | 13.16 | <0.001 |
| Attended | 23 (88.5) | 7 (36.8) | | |
| Did not attend | 3 (11.5) | 12 (63.2) | | |
| Total | 26 | 19 | | |
| *Course subject* | | | 6.253 | <0.01 |
| Animal management | 18 (69.2) | 6 (31.6) | | |
| Animal science | 8 (31.8) | 13 (68.4) | | |
| Total | 26 | 19 | | |
| *Course level* | | | | |
| FdSc | 3 (11.5) | 5 (26.3) | 1.640 | *N.S.* |
| BSc (Hons) | 23 (88.5) | 14 (73.7) | | |
| Total | 26 | 19 | | |

which equates to doctorate level studies. The minimum level of mathematics to which this cohort has been trained is grade C at GCSE level, as per institutional entry requirement.

In order to investigate student confidence in mathematics, an online questionnaire was set up and a direct link was emailed to all students in the cohort (cohort size = 101 students, 45 questionnaires were returned). The questionnaire consisted of an introduction explaining the purpose of the study, the role of the staff undertaking the research project and assurance that the survey would be anonymous. The initial section of the questionnaire included demographic information about the respondent and their previous academic qualifications. The section on confidence in mathematics was retrospective and contained sliding-scale questions on a scale of 1–10. The final section contained questions regarding feedback on the mathematics tutorial programme and reasons for either undertaking the tutorials or not undertaking them.

In addition to the survey, three 30 min focus group discussions were held with 10–12 students each in order to further investigate student feedback on the mathematics tutorials and student confidence and motivation. The focus groups contained both students who had and those who had not attended the maths tutorials. Students received a monetary incentive for participating.

Finally, the potential effect of the mathematics tutorials on student performance was analysed by applying a diagnostic test to the entire cohort at the beginning mathematics tutorial programme and to the participating students at the end of the programme. The maths tutorial programme consisted of 12 one-hour sessions delivered by an independent mathematics tutor. Both the tests and the tutorials addressed basic numeracy, e.g., multiplication, division, use of percentages and fractions, adding up and simple algebra such as rearranging equations. Thirty students followed the entire 12 session
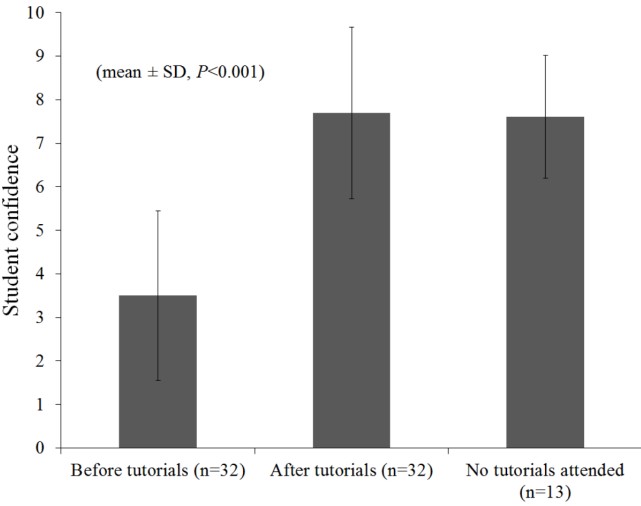

**Figure 1 Student confidence levels on a scale of 0–10.** Student retrospective self reported confidence levels on a scale of 0–10 (10 being highest) before and after attending maths tutorials ($n = 32$) compared to students who did not attend tutorials ($n = 13$).

programme. The outcomes of these tests were analysed and correlations sought with student attendance in tutorial sessions and the results for a formal maths and statistics exam.

Data were imported into Microsoft Excel (version 2007, Microsoft Inc., Redmond, WA). Statistical analyses were performed with the IBM SPSS 19 statistics suite (IBM Corporation, Armonk, NY). Bivariate analysis of the survey data was performed using Fisher's exact test or Chi-square tests. Student diagnostic test performance was analysed using Student's $T$-test, whereas student performance in the maths and statistics exam was analysed using a $T$-test for matched groups of students based on previous mathematics experience and tutorial attendance. Outcomes of the focus group discussions were grouped into themes to provide a general feedback model complementing the quantitative data as suggested by *Gibbs (1997)* and *Grudens-Schuck, Lundy Allen & Larson (2004)*.

This project was approved by the Writtle College Ethics Committee on 18 April 2012.

## RESULTS

As can be found in Fig. 1, students' retrospectively self-assessed confidence in mathematics on a scale of 0–10 was significantly improved from $3.5 \pm 0.345$ to $7.6 \pm 0.348$ by attending mathematics tutorials (mean $\pm$ S.E., $t(29) = -9.370, P < 0.001$) Students who completed all tutorial sessions, scored significantly higher in their mathematics exam ($64.3\% \pm 3.53$) compared to matched control students with similar previous maths experience who did not complete or did not attend the tutorial programme ($55.8\% \pm 2.25$) (mean $\pm$ S.E., $t(32.5) = 2.034, P \le 0.05$) (Fig. 2). The majority of students (78%) participating in the tutorial programme scored higher in their post-tutorial diagnostic test than in their pre-tutorial diagnostic test. The group score for the post-tutorial diagnostic test ($48.9\% \pm 7.3$) was significantly higher than the group score for the pre-tutorial test ($27.8\% \pm 5.5\%$) (Fig. 3).

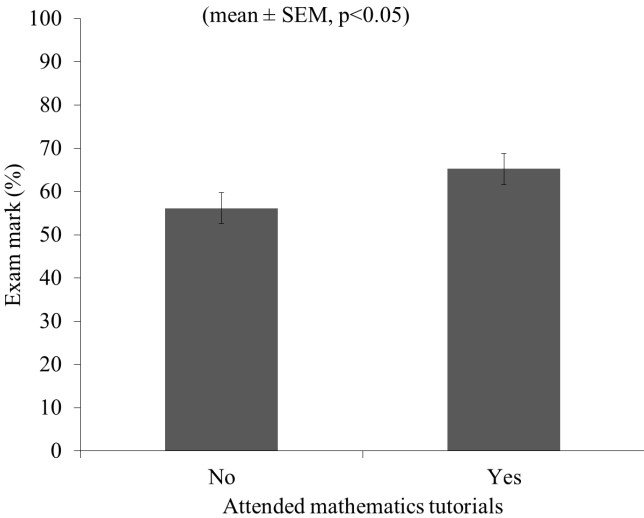

**Figure 2 Effect of attending tutorials on first year statistics exam marks (%).** Exam marks for first year maths and statistics exam for students who attended ($n = 30$) and students who did not attend the maths tutorials ($n = 11$).

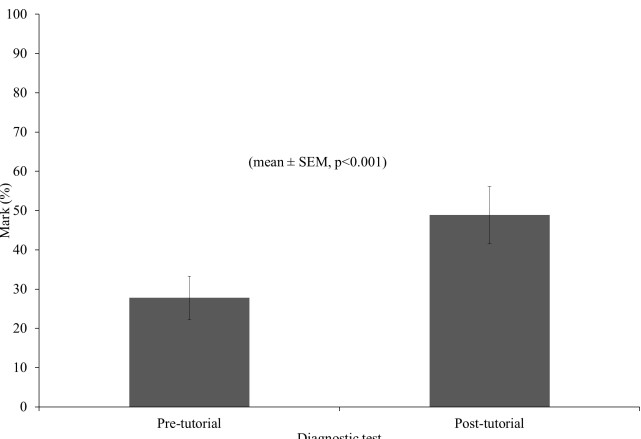

**Figure 3 Diagnostic test scores.** Pre-tutorial and post-tutorial diagnostic test scores for students who attended the maths tutorial programme ($n = 30$).

Students enrolled on an Animal Management course were more likely to only have compulsory maths experience, whereas students enrolled on an Animal Science course were more likely to have post-compulsory mathematics experience, such as A levels or International Baccalaureate ($\chi^2(1) = 6.253, P = 0.014$) (Table 1). Additionally, there was a significant association between course subject (animal management or science) and type of previous education (vocational or academic), where students enrolled on an animal management course were more likely to have a vocational background ($\chi^2(1) = 4.683$, $P < 0.05$). Furthermore, there was a significant association between students attending the mathematics tutorial service and whether or not they had post compulsory mathematics experience ($\chi^2(1) = 13.16, P < 0.001$). There was no significant association between

previous mathematics experience and the level of the course students are enrolled on ($\chi^2(1) = 1.640$).

In the group attending the initial support tutorials, mathematics confidence was significantly higher on a 10 point scale in students with post compulsory mathematics experience ($4.9 \pm 0.67$) than confidence in students with only compulsory mathematics experience ($3.1 \pm 0.37$) (mean $\pm$ S.E., $t(28) = -2.263$, $P \leq 0.05$). However, after attending the mathematics tutorials, the confidence levels between both groups were not significantly different anymore ($7.3 \pm 0.42$ and $8.7 \pm 0.36$ respectively, mean $\pm$ S.E., $t(28) = -1.839$).

In the group of students who did not attend the tutorials, the difference in mathematics confidence between students with only compulsory mathematics experience ($3.7 \pm 1.2$) and post-compulsory mathematics experience ($8.1 \pm 0.36$) was highly significant (mean $\pm$ S.E., $t(13) = -4.877$, $P < 0.001$). Additionally, non-attending students ($N = 13$) who reported they were confident in mathematics as the reason for not attending the tutorials had a significantly higher level of mathematics confidence ($8.0 \pm 0.39$) than students who gave other reasons ($4.0 \pm 1.5$) (mean $\pm$ S.E., $t(13) = 3.832$, $P < 0.01$).

The cohort of students contained a wide spread of qualifications, with the majority having completed a vocational level 3 course (e.g., Extended Diploma in Animal Management), a more academic level 3 course (A-level of IB Diploma) or a combination of the two.

The composition of the questionnaire population ($n = 45$) was a good representation of the composition of the actual student cohort ($N = 101$). Chi-square analysis revealed no tendency for gender, course level or course subject to be over or under represented in the questionnaire population (see Table 2). However, there was a slight overrepresentation of students from an FE background in the questionnaire population.

Thirty-four out of 101 students (33.7%) participated in the focus group discussions. The focus groups showed a fair representation of all courses. The feedback given by the students in the focus groups could be separated in a number of themes. These themes addressed the level of mathematics required and provided (1), relevance to the students' course (2), timing of the tutorial service (3) and improvements that could be made to the tutorial service (4).

Theme 1: Students were generally of the opinion that the level of mathematics support provided was good. They thought that the low entry level requirement supported students that struggled with basic concepts, but that more able students had the opportunity to work more independently to their own level. Some students would like to have seen more advanced mathematics addressed, but the general consensus was that this is not essential.

Theme 2: Students thought the material covered in the tutorials was generally very relevant to their course. However, in the non-attending group, students with low confidence indicated that the tutorials did not match their needs or did not fit in their schedule.

Theme 3: Student opinion was divided on the timing of the maths tutorials. A number of students would have liked to have the support during the first semester instead of the

**Table 2 Population composition.** Composition of the questionnaire sample population ($n = 45$) and the student cohort ($n = 101$). Return rate was 44.5%. Chi-square analysis revealed no over or under-representation of gender, course level or course subject, but a slight over-representation of Further Education entrants in the sample population.

| Demographic | Sample composition $n$ (%) | | $\chi^2$ | $P$ |
|---|---|---|---|---|
| | Questionnaire | Cohort | | |
| *Gender* | | | 2.177 | 0.203 |
| Male | 7 (15.6) | 27 (26.7) | | |
| Female | 38 (84.4) | 74 (73.3) | | |
| Total | 45 | 101 | | |
| *Course level* | | | 0.082 | 0.824 |
| FdSc | 8 (17.8) | 20 (19.8) | | |
| BSc (Hons) | 37 (82.2) | 81 (80.2) | | |
| Total | 45 | 101 | | |
| *Course subject* | | | 0 | 1.000 |
| Animal management | 24 (53.3) | 54 (53.5) | | |
| Animal science | 21 (46.7) | 47 (46.5) | | |
| Total | 45 | 101 | | |
| *Entry qualification level*[*] | | | 6.708 | 0.035 |
| Level 3 (FE) | 11 (24.4) | 47 (46.5) | | |
| Level 3 (A-level) | 27 (60.0) | 40 (39.3) | | |
| Other | 7 (15.6) | 14 (13.9) | | |
| Total | 45 | 101 | | |

**Notes.**

[*] "Level 3 Further Education" courses, "Access to Higher Education" courses, mixed level 3 qualifications and other types of level 3 qualifications are combined into one category "Level 3 (FE)" after consulting the "Access to Higher Education Diploma Guidelines for HE staff" published by the Quality Assurance Agency Higher Education and the "Universities Central Admissions System (UCAS) Tariff Points table" published by UCAS. A-level courses are combined with International Baccalaureate Diploma courses based on the "UCAS Tariff Points table" published by UCAS. "Other" contains level 2 and level 4–6 entrants.

second, with roughly the other half of the students of the opinion that the timing was good, as it allowed them to realise they needed help.

Theme 4: In general, students were very satisfied with the mathematics support tutorials. There were however a number of ideas raised by students which in their opinion could make the service even better. Students would like to see online support for the tutorial service, preferably in the form of online tests and revision material. Also, students would like to see the tutorial programme set up as a "drop in" surgery, instead of a 12-session long programme. Although there were one or two students who would like to see smaller groups, the consensus was that the current group size (10–12 students per session) was suitable.

## DISCUSSION

In the present study, it was clear that students with post-compulsory mathematics experience were more confident in their maths abilities than students without this experience. This may be linked to declining standards for mathematics education making

GCSE level maths not sufficient for HE bioscience requirements (*Tariq, Stevenson & Roper, 2005*; *Koenig, 2011*). However, similar criticisms exist for the current A-level maths curriculum, which means there must be other reasons. In fact, the decline in numeracy is a highly multi-factorial issue (*Tariq et al., 2010*), which makes addressing this issue challenging. *Hammouri (2004)* reported that students with a positive attitude towards mathematics tend to struggle less with the subject. As mathematically confident students are more likely to have a positive attitude towards mathematics and positive attitudes lead to better performance, raising student confidence is a good way of improving students' numeracy skills and academic performance, which is in line with *Tariq (2008)*.

In general, students indicated that they felt more confident after attending the mathematics tutorials than before, with their confidence score more than doubling and the difference between attending students and non-attending students had disappeared (Fig. 2). This indicates that small group tutorials can be an effective method of improving student maths confidence. Additionally, the students that completed the tutorial programme scored significantly higher in their post-tutorial diagnostic test when compared to their pre-tutorial diagnostic test. Also, these students scored higher in their mathematics exam than students who did not attend or complete the tutorials. This might indicate that small group tutorials can be a method of improving academic performance. However, other factors, like class attendance and attitude towards study, would also have had an influence. As such, a direct relationship between attending small group maths tutorials and academic performance cannot automatically be assumed. Nonetheless, the general usefulness of small group teaching has previously been reported by *Gunn (2007)*, and *Searl (1985)* and *MacGillivray (2009)* have previously described the use of small group tutorials for mathematical support as beneficial. Therefore, small group tutorials as maths support can still be a useful strategy to increase performance for those students who need it.

The students who did not attend the tutorial sessions because they indicated they were confident in mathematics did have significantly higher confidence scores. These students however, also had post-compulsory mathematics experience, whereas the students who did not attend tutorials but gave other reasons tended to have compulsory experience only. This indicates that there are students that do not benefit from the current programme, but who might need it.

In line with a previous report by *Koenig (2011)*, the cohort of students in this study mainly had a GCSE mathematics background. This reflects the current College entry requirements guideline where a student needs a minimum of a GCSE grade C in order to enrol on an animal science or management course. This guideline places Writtle College in line with other institutions in the UK, of which the majority (92%) requires a grade C or higher (*Koenig, 2011*).

As the animal industry is a relatively vocational industry, animal science and animal management courses by nature attract a larger number of students with a vocational background than other biosciences. This is reflected in the current study, where students with a vocational background make up around half of the cohort. In order to have access to HE Animal Science or Animal Management with a vocational qualification, the College

requires 240 UCAS points, which generally reflects a Level 3 Extended Diploma or equivalent. Nationally, the mathematics requirement for these qualifications is a GCSE grade C. As such, GCSE mathematics is common in animal sector students, even though students with this level of maths experience lack important skills (*Tariq, 2002*).

The results reflect that Animal Management students were more likely to only have compulsory mathematics experience (GCSE only), whereas Animal Science students were more likely to have post-compulsory mathematics experience. Additionally, Animal Management students were more likely to come from a vocational background whereas Animal Science students were more likely to come from a more academic background. Currently, the most common level 3 vocational course in the animal sector is the Extended Diploma in Animal Management, which might explain why students with a vocational background opt for an Animal Management related HE course. However, due to lack of research in this area, it is not possible to pinpoint the exact reasons for this phenomenon.

The feedback given by students in focus group discussions was generally very positive. The majority of the students participating in the focus groups found the tutorial programmes very helpful and saw the benefit of attending. There were however a number of suggestions made by the students which reflect a change from students as learners to students as customers in an online society. In the current tutorial programme there is no online support material available. Over half of the students indicated they would like to have the option of e-learning. *Tariq & Jackson (2008)* previously reported "Biomathtutor", a multimedia e-learning resource, to be a useful new approach to mathematics support. Offering students a blended learning experience by combining online support with small group tutorials is a concept that would meet the demands of modern day Higher Education practice (*Vasileiou, 2009*).

## CONCLUSION

Small group tutorials are an effective method of mathematics support to enhance student mathematics confidence, performance and ultimately employability, However, in a fast changing and increasingly digital HE environment, additional support in the form of e-learning might benefit those students that prefer this form of learning.

## ACKNOWLEDGEMENTS

The authors would like to thank Mr. Rob Bennet and Miss Sophia Pereira for their valuable assistance in this project.

### Funding

This project was funded by a grant from the 2012 Writtle College Learning and Teaching Fund. The funders had no role in study design, data collection and analysis, decision to publish, or preparation of the manuscript.

## Grant Disclosures

The following grant information was disclosed by the authors:
2012 Writtle College Learning and Teaching Fund.

## Competing Interests

The authors declare there are no competing interests.

## Author Contributions

- Nieky van Veggel conceived and designed the experiments, performed the experiments, analyzed the data, contributed reagents/materials/analysis tools, wrote the paper, prepared figures and/or tables, reviewed drafts of the paper.
- Jonathan Amory conceived and designed the experiments, performed the experiments, analyzed the data, contributed reagents/materials/analysis tools, reviewed drafts of the paper.

## Human Ethics

The following information was supplied relating to ethical approvals (i.e., approving body and any reference numbers):

This study was approved by the Writtle College Ethics and Animal Welfare Committee on 18 April 2012 under descriptor "Maths support for HE students".

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
