# Peer review of "The impact of maths support tutorials on mathematics confidence and academic performance in a cohort of HE Animal Science students"

_PeerJ, doi:10.7717/peerj.463_

## Round 0.1 · original submission · Major Revisions

· Academic Editor

Major Revisions

Thank you for submitting a manuscript that contributes to the important goal of evidence-based teaching in Higher Education. I have to confess that this is not my subject area and so I also thank the 2 specialist reviewers for their comments. I agree with Reviewer 2 that the research question requires clarification. Reviewer 1 raises the important point that there appears to be no baseline assessment of student ability. My own reading of your manuscript (see detailed comments below) suggests you may have information that could partly address this, but that it does not currently appear. I hope that you can include this in a substantially revised version. I think there is merit in Reviewer 1's suggestion that you could take account of the comments received and conduct further studies with a second student cohort before resubmitting. This would greatly strengthen the paper, increasing the likelihood of subsequent citations.

My detailed comments appear below:

Your focus is very much on the UK, when PeerJ is an international journal. Your introduction should be set within this wider context e.g. without assuming that all readers will be familiar with acronyms such as BBSRC, GCSE etc.
Line 42 is not fact, but opinion (albeit backed up by references)
Line 46 – by UK HE institutions? Or in other countries too?
Line 64 – clearer explanation needed of stage at which compulsory maths ends (GCSE), and a reference needed to clarify how NQF/QCF level 3 fit within qualification frameworks in the UK
Line 73 – the actual questions asked can be lodged as supplementary material
Line 74-75 – the questions on confidence were given after the tutorials had taken place, so information on confidence prior to tutorials is retrospective – this needs to be acknowledged.
Line 76 – focus groups were held only with the students that had attended tutorials?
Line 79-80 – the diagnostic tests were given only to the students that attended tutorials? If so, there is no control for other routes of improvement over the same time period e.g. growing student maturity, general increase in experience with tests or college environment, possible exposure to maths as part of other subjects being studied.
Line 79 – was the content of the tests similar to the content of the tutorials? Did the tutorials include any direct instruction in statistics?
Fig 1 – not clear how many questions were asked to establish “confidence”, or nature of these questions. Did results vary for different questions?
Fig 1 – the low confidence scores for the students that attended tutorials are retrospective – students may not remember their own pre-tutorial confidence levels accurately. This has to be acknowledged.
Fig 2- this is a rather basic analysis – could it be refined by included other factors within the No and Yes group (e.g. prior maths experience level).
Line 84 – the diagnostic tests (tests of ability) given before and after the tutorial programme could be used to answer (partially) one of the main criticisms of Reviewer 1. However, although the methods state that “ the outcomes of these tests was (sic) analysed and correlations sought with student attendance in statistics lectures and the results for a formal statistics exam” I cannot find any reference to these analyses in the Results section.
Line 134 – how did the profile of the students who participated in the focus groups (for financial reward) fit with the overall profile of the students enrolled on the different Fd and BSc degrees?

·

Basic reporting

This paper looks at the important topic of the impact of insufficient maths background for students in biology and related disciplines. In fact it is actually a small study on two courses within a single institution but as such adds to the literature. My background is in statistics and I have taught maths/stats to non-mathematicians but would not be an expert on 'research' in this area but there appear to be references to several papers in the field. The article also confirms to the required style.The figures are OK but personally I'd prefer to see 2 or 3 means and standard errors put as a table and not as a bar graph and also I'd like to see the sample sizes for each group included. The submission is certainly self-contained and contains lots of statistical tests

Experimental design

I do have a slight issue with the experimental design. One 'finding' and perhaps what the reader would most be interested in is do the maths tutorials work. The authors can show that the tutorials modify confidence and that is fine but the ability question CANNOT be answered in this study as it is missing a baseline measure of ability. The fact that the final year stats mark was larger in the group attending maths tutorials does NOT definitely imply that the tutorials work and the discussion statements here are too strong. The groups are self selecting and it might be that the most able within the background category students are deciding to take tutorials - certainly from personal experience there would be a strong correlation between turning up to classes and performance. Without giving the students a test of ability prior to the tutorials it is impossible to separate out effect of tutorials from different baseline ability.I would also like to see sample sizes given more prominence. The reader has to hunt to find how big the potential population is, how big the sample is at each stage and what the selection mechanism is i.e. I expect it is willingness to participate.
It was good however to check representativeness of the sample.

Validity of the findings

The statistical tests appear to be valid. The statistical problems I have identified are more regarding design and impact on effect of tutorials on attainment as detailed above. I am not a qualitative researcher so don't feel able to comment much on focus groups apart from they seem here to indicate some opinions about the tutorials. It would be good to have some sense of strength of opinion i.e. are these one voice or a consensus of the 34 students (out of 101). I also struggle with sample sizes as at points the number 101 and then the number 146 are mentioned and so some clarity would be helpful.I believe the discussion in places overplays the importance of findings and could be toned down a little.

Additional comments

Aside from my comments in the three boxes above there are some typos and mathematics I would consider personally as singular in the abstract. I would personally suggest either a resubmitted article that acknowledges the weakness of the design for attributing exam success to tutorials but focussing on confidence or perhaps running the experiment again with the next cohort and using this as pilot data and this time including a baseline assessment of maths ability i.e. a test on entry.

Reviewer 2 ·

Basic reporting

The authors make the very valid case that the biosciences are becoming increasingly quantitative and it is important that students are sufficiently well-grounded in quantitative techniques. Unfortunately insufficient numbers coming into the system have such grounding and as is now well-known additional measures must be put in place to support them if they are to succeed. The approach described in this paper is to offer small-group mathematics tutorials (12 one-hour sessions) for students of Animal Management/Science. The paper reports on the efficacy of this intervention.

The text is clear and unambiguous. Appropriate background information is provided.

Experimental design

The research question is not explicitly stated and the authors should review this aspect.

In order to evaluate the efficacy of the tutorials, research was undertaken through a mixed quantitative/qualitative approach using survey, focus groups and analysis of academic performance. Ethical approval was gained.

Validity of the findings

Conclusions are appropriate and clearly stated.

Additional comments

This is interesting reading. I have a few suggestions for consideration by the authors:

1. Review the requirement to specify the research question.

2. It would be helpful for the reader to know a little more about the small-group tutorials: in what sense are they optional/compulsory ? how are they encouraged/promoted/advertised ?

2. Line 68: It would be helpful to be told the number in the cohort without having to try to ascertain this information from Table 2. In fact, additional information about cohort size, number responding to the questionnaire, etc would be helpful here. I had difficulty trying to unpick this.

3. I wondered whether Figure 1 might be improved. On the face of it, the confidence of those who did not attend tutorials is in line with those who did once the intervention was over. However, this masks the fact that some who did not attend were very confident and others were not.

---

## Round 0.2 · accepted · Accept

· Academic Editor

Accept

Thank you for a revised manuscript that addressed carefully the previous comments of the reviewers and of the editor.